# Cryopreservation of Pig Semen Using a Quercetin-Supplemented Freezing Extender

**DOI:** 10.3390/life12081155

**Published:** 2022-07-29

**Authors:** Seonggyu Bang, Bereket Molla Tanga, Xun Fang, Gyeonghwan Seong, Islam M. Saadeldin, Ahmad Yar Qamar, Sanghoon Lee, Keun-Jung Kim, Yun-Jae Park, Abdelbagi Hamad Talha Nabeel, Il-jeoung Yu, Akila Cooray, Kyu Pil Lee, Jongki Cho

**Affiliations:** 1Laboratory of Theriogenology, College of Veterinary Medicine, Chungnam National University, Daejeon 34134, Korea; bangsk97@o.cnu.ac.kr (S.B.); tanga@o.cnu.ac.kr (B.M.T.); fx2442@o.cnu.ac.kr (X.F.); 202050377@o.cnu.ac.kr (G.S.); islamms@cnu.ac.kr (I.M.S.); sanghoon@cnu.ac.kr (S.L.); 2Research Institute of Veterinary Medicine, Chungnam National University, Daejeon 34134, Korea; 3Department of Physiology, Faculty of Veterinary Medicine, Zagazig University, Zagazig 44519, Egypt; 4Collage of Veterinary and Animal Science, University of Veterinary and Animal Sciences, Lahore 54000, Pakistan; ahmad.qamar@uvas.edu.pk; 5Livestock Experiment Institute, Government of Chungcheongnam-do, Cheongyang-gun 33303, Korea; imkjkim@korea.kr (K.-J.K.); dso446@korea.kr (Y.-J.P.); 6Laboratory of Theriogenology and Reproductive Biotechnology, College of Veterinary Medicine and Bio-Safety Research Institute, Jeonbuk National University, Iksan 54596, Korea; nabeeltalha@yahoo.com (A.H.T.N.); iyu@jbnu.ac.kr (I.-j.Y.); 7Department of Physiology, College of Veterinary Medicine, Chungnam National University, Daejeon 34134, Korea; akiladushyantha@gmail.com (A.C.); kplee@cnu.ac.kr (K.P.L.)

**Keywords:** oxidative damage, cryopreservation, pig sperm, quercetin

## Abstract

Reactive oxygen species (ROS) produced during freeze–thaw procedures cause oxidative damage to the sperm, reducing fertility. We aimed to improve the post-thaw quality of pig sperm by quercetin (QRN) supplementation to reduce the cryodamage associated with the freeze–thaw procedure. Four equal aliquots of pooled boar semen were diluted with a freezing extender supplemented with different concentrations of QRN (0, 25, 50, and 100 µM) and then were subjected to cryopreservation in liquid nitrogen. Semen analysis was performed following 7 days of cryopreservation. Results demonstrated that the semen samples supplemented with 50 µM QRN significantly improved the post-thaw sperm quality than those subjected to other supplementations (*p* < 0.05). Semen samples supplemented with 50 µM QRN showed significantly improved plasma membrane functional integrity (47.5 ± 1.4 vs. 43.1 ± 4.1, 45.3 ± 1.7, and 44.1 ± 1.4) and acrosome integrity (73.6 ± 3.4 vs. 66.3 ± 2.4, 66.7 ± 3.6, and 68.3 ± 32.9) as compared to the control, 25 µM, and 100 µM QRN groups, respectively. The mitochondrial activity of the 50 µM QRN group was greater than control and 25 µM QRN groups (43.0 ± 1.0 vs. 39.1 ± 0.9 and 41.9 ± 1.0) but showed no difference with the 100 µM QRN group. Moreover, the 50 µM QRN group showed a higher sperm number displaced to 1 cm and 3 cm points in the artificial mucus than other groups. Therefore, supplementing the freezing extender with QRN can serve as an effective tool to reduce the magnitude of oxidative damage associated with sperm freezing.

## 1. Introduction

Artificial insemination using frozen pig semen still has a low success rate due to inefficient freeze–thaw procedures [1]. The main reasons for inefficient freezing include the sensitivity of pig sperm to low temperatures and intolerance to oxidative stress (OS) [2]. OS associated with freeze–thaw is due to an enhanced reactive oxygen species (ROS) production [3] that severely damages different cellular components including lipids, proteins, carbohydrates, and nucleic acids [4]. Consequently, results include infertility issues due to the reduced sperm motility and interference in sperm–oocyte interaction [5]. Previous reports have demonstrated the importance of antioxidants during cryopreservation of boar semen, such as tocopherols [6,7], reduced glutathione [8,9], and a combination of lutein, Trolox, and ascorbic acid [10]. QRN has been used for semen cryopreservation in humans and different animal species and has shown beneficial effects on sperm characteristics and functions after sperm freezing and thawing [11,12,13,14].

Higher contents of polyunsaturated fatty acids along with a reduced amount of cholesterol in pig sperm make them more susceptible to oxidative damage [15,16]. Recent reports have indicated the presence of a complex redox system that combines the antioxidative capability of seminal plasma and sperm [17,18]. However, the seminal plasma of pigs has a low antioxidant capacity [16]. Moreover, the removal of seminal plasma before the freezing of pig sperm affects the fluidity of the plasma membrane, resulting in partial loss of protection [19].

Strategies using antioxidants have been introduced to overcome the drawbacks associated with sperm freezing in pigs. Alpha-tocopherol has shown high competence in cryopreserved boar sperm [20]. Similarly, glutathione has been added as a supplement to reduce free radicals during the freeze–thaw procedure [21,22], and a combination of lutein, Trolox, and ascorbic acid improved sperm characteristics after cryopreservation [23]. Quercetin (QRN) (2-(3,4-dihydroxyphenyl)-3,5,7-trihydroxy-4H-chromen-4-one) is a flavonoid found in a variety of fruits and vegetables [24]. QRN has antioxidant properties; it can efficiently scavenge free radicals and chelate metal ions [25,26]. Moreover, QRN has been used for freezing semen of several farm animal species and in humans [11,12,13,14,27,28,29,30,31]. QRN possesses antioxidative and free radical scavenging properties, as revealed by several in vitro and in vivo studies [11,27,32,33,34,35,36]. Only a few studies have focused on the effects of QRN supplementation during the thawing process.

In pigs, the supplementation of a sperm freezing extender with QRN has not been studied yet. Therefore, the current study was aimed to investigate the supporting effects of different levels of QRN on sperm characteristics, functions, apoptosis, ROS neutralization, and essential transcripts expression during the freezing of pig sperm.

## 2. Materials and Methods

### 2.1. Chemicals

All chemicals were purchased from Sigma-Aldrich (St. Louis, MO, USA) unless otherwise stated.

### 2.2. Semen Collection and Handling

Semen samples were collected twice a week for 4 weeks from four Yorkshire boars, (i.e., total of 32 individual doses in eight pools). Semen samples were provided by the Livestock Experiment Institute, Government of Chungcheongnam-do, and moved to the laboratory at 17 °C within 2 h of collection. Semen samples with ≥70% motility, ≥80% viability, and a sperm concentration ≥100 × 10^6^ sperm cells/mL were pooled and processed. Semen processing and analysis were replicated 8 times for each group.

### 2.3. Ethical Approval

All experimental procedures were conducted in accordance with the guidelines for the management of laboratory animals at Chungnam National University (Approval No. 202103-CNU-079).

### 2.4. Safety and Toxicity of Dimethyl Sulfoxide (DMSO) for Sperms

Because QRN is soluble in DMSO, toxicity evaluation of DMSO was first performed. Each sperm-freezing medium was treated with 0, 0.5, 1.0, 1.5, and 2.0% DMSO, and then, freezing was initiated. After that, the sperm straws were thawed, and CASA (Sperm Class Analyzer^®^ CASA system, MICROPTIC, Nikon Ci-L, Tokyo, Japan) was performed.

### 2.5. Semen Cryopreservation

Sperm freezing was performed according to Kim et al. [37]. In brief, pooled semen samples were washed using centrifugation at 500 g with basic diluent supplemented with D(+)-Glucose (6.00 g/L), EDTA (0.45g/L), sodium citrate (1.38g/L), sodium bicarbonate (0.2g/L), Trizma base (1.00 g/L), citric acid (0.5g/L), cysteine (0.01g/L), BSA (0.8g/L), and kanamycin sulfate (0.06g/L). The washed samples were adjusted to 200 × 10^6^ sperms per ml with LEY extender supplemented with lactose solution (310 mM) (80%), egg yolk (20%), and kanamycin sulfate (100 µg/mL). Thereafter, the quercetin was dissolved in DMSO to make a 10 mM stock. Finally, the semen suspension was divided and diluted with a freezing extender (LEY extender 89.5%, glycerol 9%, orvus ES paste 1.5%, trehalose 100 mM) supplemented with 25, 50, or 100 µM QRN or without QRN (control). Following a multistep dilution process, the semen was diluted to a final concentration of 100 × 10^6^ cells/mL. The diluted semen was placed in 0.5 mL semen straws (Minitube GmbH, Ref. 13408/0010, Tiefenbach, Germany) and equilibrated at 4°C for 45–60 min. The semen straws were frozen by placing them horizontally 2 cm above the surface of liquid nitrogen (LN_2_) for 15 min. The sperm straws were then placed in liquid nitrogen and stored.

### 2.6. Post-Thaw Semen Analysis

Following a week of semen cryopreservation, frozen semen samples were thawed in a water bath at 37 °C for 30 s [38]. Post thaw, sperm were diluted to a concentration of 10 × 10^6^/mL and used for analysis. To evaluate the kinematic parameters of post-thawed sperm, the CASA software imaging system (Sperm Class Analyzer^®^ CASA system, MICROPTIC, Nikon Ci-L, Tokyo, Japan) was used. The parameter settings for the CASA software were 25 frames per one second (25 Hz). The minimum and maximum areas of the detected objects were 10 μm^2^ and 80 μm^2^, respectively. In addition, Standard Count 8 Chamber Slide—20 µm (Leja^®^, Nieuw-Vennep, The Netherlands) was used to analyze the kinematic parameters of sperm. In brief, sperm samples (5 µL) were placed on a counting slide for assessment, and for each sample, 5 different fields were randomly examined. At least 200 sperm were tracked for 1 s at 25 Hz for each sample. The kinematic parameters analyzed included the percentage of motile sperm, progressive motility, curvilinear velocity (VCL), average path velocity (VAP), straight-line velocity (VSL), straightness, linearity, and amplitude of lateral head displacement (ALH). Each sample’s analysis was repeated 8 times to increase measurement precision.

#### 2.6.1. Assessment of Sperm Plasma Membrane Functional Integrity

Plasma membrane functional integrity was analyzed using the hypo-osmotic swelling (HOS) assay using 190 mOsm sucrose solution [39]. Each group received one drop (50 µL) of post-thaw semen, which was mixed with 0.5 mL of HOS solution and incubated at 37 °C for 30 min. The incubated mixture was placed on a pre-warmed glass slide, and 200 sperm per sample were evaluated for their ability to expand using a phase-contrast microscope within 5–10 min (Eclipse Ts2, Nikon, Tokyo, Japan). Sperm with intact plasma membrane showed head swelling and coiling of the sperm tail.

#### 2.6.2. Mitochondrial Activity Assay and Acrosome Integrity

Mitochondrial activity of post-thaw pig sperm was assessed using a combination of two fluorescence stains, namely rhodamine 123 (R123) and propidium iodide (PI) [40]. For storage, 30 µL of R123 solution (5 mg/mL in Me_2_SO) was diluted with 120 µL of Me_2_SO and split into 30 µL aliquots. After thawing the sperm samples, they were diluted with buffer 1 to a concentration of 20 × 10^6^ sperm/mL, and 3 µL of R123 working solution was added. The sperm stain suspension was incubated in the dark for 15 min at 37 °C. After incubation, sperm samples were treated with 10 µL of PI solution (0.5 mg/mL in PBS) and incubated for another 10 min at 37.8 °C. The sperm pellet was spun at 500 × g for 5 min and resuspended in 1 mL of PBS. A drop of the sperm suspension (10 µL) was put on a microscopic slide and covered with a coverslip. The functionally active mitochondria were identified by the presence of deep green fluorescence at the midpiece of the sperm. The integrity of the acrosome in post-thaw sperm samples was assessed using fluorescein isothiocyanate-conjugated peanut agglutinin (FITC-PNA) staining [41]. The acrosomal status was monitored using an epifluorescence microscope (×1000 magnification; Eclipse Ts 2, Nikon, Tokyo, Japan). At least 200 sperm per smear were examined and classified as intact acrosome membrane (strong green fluorescence) or damaged acrosomal membrane (no fluorescence).

#### 2.6.3. Mucus Penetration Test

Mucus penetration tests were performed using artificial mucus (modified synthetic oviductal fluid) [42]. The fluid was loaded into sealed flat capillary tubes (80 ± 0.5 mm long, 1.25 ± 0.05 mm wide, Hilgenberg GMBH, Stutzerbach, Germany). The capillary tubes were placed vertically for 15 min to check the seal’s tightness and remove bubbles before being inserted into an Eppendorf tube containing 100 µL of semen suspension and placed vertically for 2 h at room temperature (25–28 °C). The number of sperm that reached the 1 and 3 cm marks in the capillary tube were counted as sperm quality indicators.

#### 2.6.4. Assessment of Reactive Oxygen Species (ROS) Level

The ROS level was determined in the method outlined by Guthrie and Welch (2007) [43]. Sperm ROS levels were assessed using 2,7-dichlorodihydrofluorescein diacetate (DCFDA, Sigma, St. Louis, MO, USA), which is used to detect H_2_O_2_ [44]. The mean fluorescence intensity of DCFDA was measured to evaluate the intracellular mean H_2_O_2_ per viable sperm population. ROS level was calculated considering only PI-negative sperm. All stained spermatozoa fluorescence signals were analyzed using a FACSCalibur^®^ flow cytometer (Becton Dickinson, Franklin Lakes, NJ, USA) coupled with a 15 mW air-cooled 488 nm argon-ion laser.

#### 2.6.5. Assessment of Viability and Apoptotic Status using Flow Cytometry

Sperm apoptosis status was evaluated through flow cytometry using differential annexin V-FITC and propidium iodide (PI) staining [45]. In brief, spermatozoa were pelleted twice in PBS at 300 × g for 5 min before being diluted in 1 mL of 1 annexin buffer (5 × 10^6^ sperm/mL). Then, 100µL of this solution was collected and combined with 5µL annexin-FITC stain and 5µL propidium iodide in fresh 1.5 mL tubes. For 15 min, the mixture was kept in the dark at room temperature (25 °C). The tubes were then filled with 400µL of 1 annexin buffer, and flow cytometry was performed. Flow cytometry analysis was performed through BD Accuri™ C6 plus (Becton Dickinson, BD Biosciences, Franklin Lakes, NJ, USA). The flow cytometer was fitted with blue (488 nm, solid-state, 20 mW) and diode red (640 nm, 14.7 mW) excitation lasers. The fluorescent probes used in this experiment were annexin V-FITC (apoptosis detection kit I (BD Biosciences), and propidium iodide was excited using a 488 nm blue-solid state laser. Live spermatozoa stained with annexin V were detected using a filter detector 533/30 BP (wavelength range 511–543 nm). The signal from dead sperm stained with propidium iodide was detected using a filter detector 586/42 BP (wavelength range 565–607 nm). The flow rate of the flow cytometer was medium (35 μL/min, 16 μm core). Sperms were categorized into four different groups according to the individual or double staining of annexin V and PI: viable non-apoptotic, early apoptotic, late apoptotic, and necrotic cells. The last group contained late apoptotic and necrotic cells, which were categorized as dead cells. Apoptotic status was calculated considering only PI-negative sperm. Sperm populations were divided into regions and quadrants. The data were analyzed using BD Accuri™ C6 Plus Flow cytometer software.

#### 2.6.6. Measurement of Sperm Absolute Membrane Potential

Firstly, the calibration plot of theoretical membrane potential and median fluorescence intensity of DiSC_3_(5) (3,3′ dipropylthiadicarbocyanine iodide) (Invitrogen, Eugene, OR, USA) was generated by time-lapse flow cytometry as detailed in Matamoros-Volante et al. (2020) [46] (Appendix A). Briefly, non-capacitated boar sperms were diluted to a final concentration of 1–3 × 10^6^ cells/mL in non-capacitating media (in mM: NaCl 90, KCl 4.68, glucose 2.78, CaCl_2_ 1.8, KH_2_PO_4_ 0.37, MgSO_4_ 0.2, sodium pyruvate 0.33, sodium lactate 21.39, HEPES (4-(2-hydroxyethyl)-1-piperazineethanesulfonic acid) 23.8, pH adjusted to 7.4 with NaOH). DiSC_3_(5) fluorescent dye, a probe sensitive to membrane potential, was used to evaluate the membrane potential of spermatozoa as means of fluorescence measurements. After 15 min incubation with 100 µM DiSC_3_(5) at 37 °C, flowcytometry (FACS) data acquisition was begun. Fluidics rate of the system was set to the slow (14 µL). Basal fluorescence of the cells was measured for the first 120 s. It was followed by the addition of 1 µM valinomycin (K+ ionophores) and sequential addition of KCl to obtain the final concentration of 5,10,15, 25, and 40 mM in the sample corresponding to the theoretical membrane potential −80, −63, −52, −40, and −28 mV accordingly. Nernst equation for equilibrium was used to calculate the theoretical membrane potentials corresponding to each final KCl concentration. The median fluorescence value of the sperm subpopulation that responds to KCl was then plotted against theoretical absolute membrane potentials to obtain a linear regression. Secondly, the semen samples treated with different quercetin concentrations were thawed and resuspended in the capacitation media (non-capacitating media + 25 mM NaHCO3 and 0.5 *w*/*v* BSA). Samples were incubated at 37 °C for 6 h to induce sperm capacitation. Afterward, the fluorochrome was loaded as previously mentioned. The DiSC_3_(5) intensity of the quercetin-treated samples (control, 25 µM, 50 µM, 100 µM) was measured, and their absolute theoretical membrane potential values were predicted using the linear regression obtained from the calibration. For each sample, the median fluorescence of a minimum of 20,000 sperms was measured.

### 2.7. Gene Expression through Real-Time PCR Analysis

TRI reagent (Invitrogen, Carlsbad, CA, USA) was used to extract total RNA from pure post-thaw sperm supplemented with or without quercetin (n = 8, each group). Complementary DNA (cDNA) was synthesized using 2X RT Pre-Mix (BIOFACT, Daejeon, Korea) with a total volume of 20 μL (4 μL of 5 × RT buffer, 1 μL of RT enzyme mix, 1 μL of oligo dT primer, 1 μg of RNA, and up to 20 μL of RNase-free dH_2_O). Quantitative real-time polymerase chain reaction (RT-qPCR) was used to analyze the expression of pro-apoptotic BCL2-associated X (*BAX*), Bcl-2 homologous antagonist/killer (*BAK*), anti-apoptotic B-cell lymphoma-like1 (*BCL-2l*), B-cell lymphoma-extra-large (*Bcl-xl*), cyclooxygenase isoenzyme type 2(*COX-2*), and phospholipase C zeta (*PLCz*) (Table 1). RNA was extracted using the TRI reagent (Invitrogen, Carlsbad, CA, USA) from post-thawed sperm cryopreserved with different concentrations of QRN or control. RNA quality was checked with Nanodrop spectrophotometry (Nanodrop 2000, Thermo Fischer, CA, USA). The expression levels of the examined genes were quantified by relative quantitative real-time PCR through the CFX96 real-time PCR detection system (Bio-Rad, Hercules, CA, USA) using SYBR 2X Real-Time PCR Pre-Mix (BIOFACT) against the housekeeping gene GAPDH through the equation R = 2 ^[DCq sample-DCq control]^.

### 2.8. Statistical Analysis

Statistical Package for Social Sciences (SPSS) version 24.0 software (IBM, Armonk, NY, USA) was used for statistical analysis among different treatments. The normality of data distribution was evaluated through Shapiro–Wilk W test. All the values are expressed as the mean ± standard error of the mean (SEM). One-way analysis of variance (ANOVA) followed by post hoc Tukey’s multiple comparison test was used to analyze the differences between the tested parameters in different treatment groups. *p* < 0.05 was used to indicate statistical significance. 

**Table 1 life-12-01155-t001:** Primer sequences used for the analysis of gene expression in post-thaw porcine sperm.

Gene	Primer Sequence (5′–3′)	Accession No.
*GAPDH*	F: AGAAGGTGGTGAAGCAGG R: AGCTTGACGAAGTGGTCG	XM_003126531
*BAX*	F: AAGCGCATTGGAGATGAACT R: CTGGACTTCCTTCGAGATCG	AJ606301
*BAK*	F: ACCGACCCAGAGATGGTCAC R: CAGTTGATGCCACTCTCGAA	AJ001204
*BCL-2l*	F: GAAACCCCTAGTGCCATCAA R: GGGACGTCAGGTCACTGAAT	NM_214285
*BCL-xl*	F: CTGAATCAGAAGCGGAAACC R: GGGACGTCAGGTCACTGAA	AF216205
*COX-2*	F: CAACGCCTCTACCAGTCTGC R: TTCGGGTGCAGTCACACTTA	ss319605207
*PLCz*	F: CATGAGATAGACTGCCCTCTGA R: CTGAATTCCCAGCAGACATTC	ss319605203

## 3. Results

In our preliminary experiments, we confirmed the safety and/or toxicity of DMSO in the cryopreservation of pig semen, and the quality of straws treated with DMSO at each concentration was evaluated post thaw. The kinematic parameters of post-thaw sperm were measured using CASA. The motility of DMSO-treated semen samples did not differ significantly between concentrations of 0%, 0.5%, 1.0%, and 1.5%. However, in the sample treated with 2.0% DMSO, motility was significantly reduced. On the other hand, there was no significant difference in progress motility among all groups (Table 2).

### 3.1. Effect of Quercetin on Sperm Motility after Freezing/Thawing

Based on improvements in post-thaw sperm quality and kinetic parameters of pig sperm supplemented with various concentrations of QRN, the optimum concentration of QRN was determined (Table 3). Sperm samples supplemented with 50 µM QRN showed the highest post-thaw motility (33.73 ± 0.85%) compared to other groups (*p* < 0.05).

### 3.2. Integrity of Plasma Membrane and Acrosome

Sperm samples supplemented with 50 µM QRN significantly improved the acrosome integrity, mitochondrial activity, and osmotic competence as well as the mucus penetration ability of frozen–thawed pig sperm (Table 4). The effect of QRN supplementation on sperm viability did not show any significant difference among different QRN concentrations but was higher than that of the control group (46.9 ± 2.6%, 47.5 ± 1.9%, 45.4 ± 2.5%, vs. 43.3 ± 1.0%, respectively). The percentage of sperm tolerant of the HOS test in the 50 µM QRN group was significantly higher (47.5± 1.4%) compared with that of the control, 25, and 100 µM QRN groups (43.1 ± 4.1%, 45.3 ± 1.7%, and 44.1 ± 1.4%, respectively; *p* < 0.05). Similarly, the percentage of sperm with intact acrosome was significantly higher in the 50 µM QRN-supplemented group (73.6 ± 3.4%) when compared with the control, 25, and 100 µM QRN groups (66.3 ± 2.4%, 66.7 ± 3.6%, and 68.3 ± 2.9 %, respectively; *p* < 0.05). However, sperm samples supplemented with 50 and 100 µM QRN groups showed greater mitochondrial activity when compared with the control and 25 µM QRN groups (43.0 ± 1.0% and 43.2 ± 1.8 % vs. 39.1 ± 0.9% and 41.9 ± 1.0%, respectively, *p* < 0.05). Supplementation of freezing extender with 50 µM QRN improved the mucus penetration ability of the frozen–thawed sperm compared with the other groups showing superior ability over the control group after challenging with 1 cm and 3 cm of mSOF medium (Table 5).

### 3.3. Impacts of Quercetin on Oxidative Stress and Apoptosis of Frozen/Thawed Boar Sperm

DCFDA sperm staining results showed that 50 µM of QRN substantially reduced the ROS level in the frozen-thawed sperm in comparison to the other experimental and control groups, indicating the potent antioxidative properties of QRN (Figure 1). Moreover, sperm samples supplemented with 50 µM QRN significantly reduced the percentage of apoptotic sperm than that of control and other groups (Figure 2).

### 3.4. Effect of Quercetin on Sperm Membrane Potential after Freezing/Thawing

Increasing quercetin concentration shows tendency for the DiSC_3_(5) and membrane potential of the spermatozoa to increase (depolarize) (Figure 3A). The theoretical membrane potential of the 100 µM quercetin-treated sperms was around −66 mV, and it was statistically significant to that of 25 µM quercetin-treated samples (Figure 3B). However, the DiSC_3_(5) median intensity and membrane potential of 25-µM- and 50-µM-treated samples were not significantly different from those of the control counterparts.

### 3.5. Effects of Quercetin on Sperm Gene Expression

We analyzed gene expression levels in post-thaw pig sperm samples supplemented with QRN. In the 50 µM QRN group, apoptosis-related genes *BAX* and *BAK* were significantly decreased, and anti-apoptosis factors *BCL-2l* and *BCL-xl* were significantly increased. In addition, there was no difference in the *COX-2* gene between the control group and the 100 µM QRN group, but there was a significant difference between the 50 µM QRN and 100 µM QRN groups. The *PLCz* gene was significantly elevated in the 50 µM QRN and 100µM QRN groups (Figure 4).

## 4. Discussion

Sperm freezing is considered an important tool of assisted animal reproduction used for preserving the genetic merits of valuable animals [29]; however, pig sperms are more sensitive to cryodamage than other domestic animal species [2,47,48].

The current results provided an ameliorative effect of QRN on the cryodamage caused by porcine semen freezing and thawing in terms of improving sperm kinematic properties, membrane and acrosomal integrity, mitochondrial activity, and mucus penetration ability. Moreover, the oxidative stress and apoptosis index were also reduced by the QRN supplementation.

Potassium channels of the mammalian spermatozoa play a vital role in sperm volume regulation and hyperpolarization of the plasma membrane, which primes the process known as capacitation [49]. Pig sperms consist of many potassium channels, and inhibition of these potassium channels affects their ability to undergo capacitation or acrosome reaction highlighting the functional importance of K+ ion channels in boar sperm physiology [50]. Previous reports of electrophysiology of human spermatozoan ion channels have shown quercetin to modulate human Slo3 (responsible for K+ ion conductance in sperms) in in vitro settings [51]. Therefore, we assessed the effect of long-term treatment of quercetin on boar spermatozoon, which is especially relevant to the potassium ion channel modulations, resulting in potential membrane changes. Hyperpolarization of the sperm membrane is known to be associated with a series of signal cascades in the capacitation of mammalian spermatozoa. Hence, we incubated the treated semen samples under capacitation conditions (non-capacitating media+ 25 mM NaHCO_3_, and 0.5% *w*/*v* BSA incubated for 6 h at 37 °C) to induce membrane hyperpolarization. Although there is a tendency for depolarization with the increase of quercetin concentration, the lower concentrations tested did not show a significantly different membrane potential compared to the control. Altogether, this suggests that lower concentrations of quercetin treatment for a long period do not impair the boar sperm K+ ion channel physiology, and the sperm’s ability to elicit membrane potential changes when undergoing the capacitation is kept intact compared to the control counterparts tested.

On the molecular level, the expression of apoptotic transcripts such as *BAK* and *BAX* were reduced, while anti-apoptotic transcripts *BCL-2l* and *BCL-xl* were elevated in the QRN-supplemented extender. The decreased *BAX* and increased *BCL2* expression demonstrated a significant correlation with improved sperm motility [52]. Antioxidants supplementation such as L-carnitine showed similar effects on the expression of *BAX* and *BCL2* in mice [53]. This result coincides with our finding in reducing the apoptotic index caused by QRN supplementation. Furthermore, *COX2* was reduced by QRN supplementation, which might indicate a better-quality sperm than that found in the control group [54,55]. Interestingly, the increased *PLCz* gene expression with QRN supplementation indicates a more functional spermatozoon, such as generating calcium ions oscillations during the oocyte activation [55,56,57].

QRN has not been previously used as a supplement for freezing media for boar sperm; however, it has been used only after thawing. Tvrda et al. [58] investigated the effects of supplementing QRN to the thawing medium (Beltsville Thawing Solution, BTS) on boar semen storage for 72 h and found a marked improvement in OS markers and mitochondrial activity in 10–25 μM QRN-supplemented samples. Thawing medium supplemented with 0.25 mM QRN for 10 h improved post-thaw sperm motility and viability, decreased polyspermy, and enhanced embryonic development after in vitro fertilization (IVF) [59]. These effects were speculated to be associated with the reduction of free radicals (hydrogen peroxide) and lipid peroxidation [59]. Similar results were also shown by Kim et al. [60], who found that supplementing the thawing medium with 1 μM for 6 h improved sperm membrane integrity, mitochondrial activity, and embryonic development after IVF. Supplementation of 0.1 mM QRN has also shown beneficial impacts on the cooling of boar sperms when added to BTS medium for 7 days [61]. The results of the current study showed a positive effect of supplementing the freezing extender with 50 µM QRN on sperm characteristics after thawing, as indicated by the improvement in osmotic resistance, acrosome integrity, and mitochondrial activity. In addition, QRN improved the mucus penetration capability of sperm and reduced ROS and apoptosis in frozen/thawed sperm. All these parameters reflected the powerful antioxidative action of QRN in neutralizing ROS [62,63,64] and protecting sperm from cryodamage associated with sperm freezing and thawing [29].

## 5. Conclusions

The results of this experiment confirm that the QRN supplementation during freezing prevents the cryodamage of sperm. The optimum concentration of QRN for the protective freeze–thaw of pig sperm against OS is 50 μM QRN compared to the other tested concentrations of QRN. At this optimal concentration, QRN improves the key fertility indicators of pig sperm, including the percentage of live sperm and the integrity of vital structures, including the plasma membrane and acrosome, as well as the gene expression of apoptosis and sperm functions-related genes. These findings require subsequent studies to investigate the pregnancy outcomes and in vitro fertilization of the porcine frozen semen.

## Figures and Tables

**Figure 1 life-12-01155-f001:**
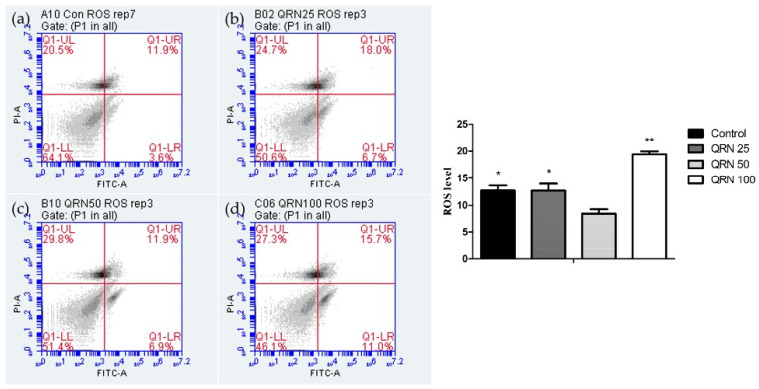
Effect of quercetin on ROS levels of post-thaw boar sperm. (**a**) Control, (**b**) QRN 25, (**c**) QRN 50, and (**d**) QRN 100. (**a**–**d**) Results of DCFDA/PI staining and ROS analysis by flow cytometry. The bar graph is the result of comparing the ROS index value using the flow cytometry result. Values carrying different asterisks (* and **) are significantly different when *p* < 0.05 (n = 8).

**Figure 2 life-12-01155-f002:**
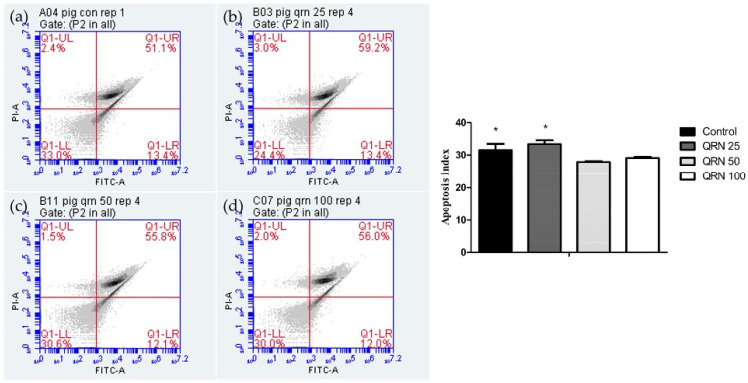
Detection of annexin V-FITC in post-thaw boar sperm. (**a**) Control, (**b**) QRN 25, (**c**) QRN 50, and (**d**) QRN 100. (**a**–**d**) Results of annexin V-FITC/PI staining and analyzed using flow cytometry. Values carrying asterisk (*) are significantly different when *p* < 0.05 (n = 8).

**Figure 3 life-12-01155-f003:**
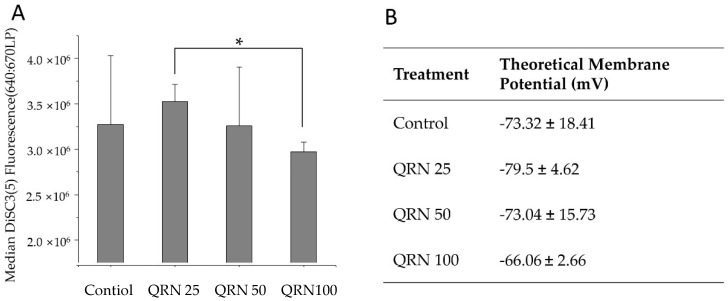
Theoretical absolute membrane potential indicated by DiSC_3_(5) fluorescence intensity. (**A**) the values of DiSC_3_(5) fluorescence intensity of control and QRN-treated groups. (**B**) Estimation of the theoretical membrane potential. Asterisk (*) indicate statistical significance when *p* < 0.05.

**Figure 4 life-12-01155-f004:**
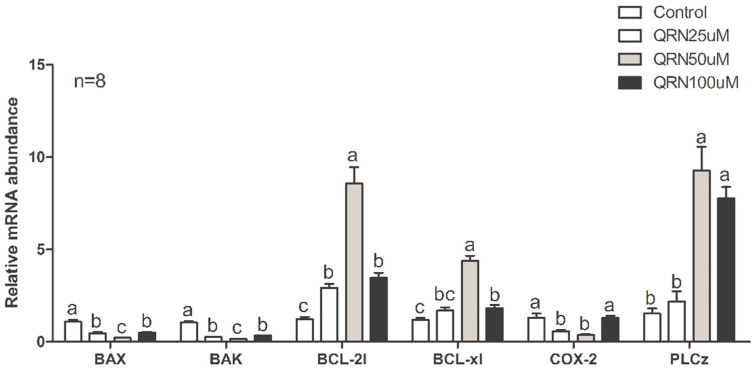
Gene expression levels of pro-apoptotic BCL2-associated X (*BAX*), Bcl-2 homologous antagonist/killer (*BAK*), anti-apoptotic B-cell lymphoma-like1 (*BCL-2l*), B-cell lymphoma-extra-large (*Bcl-xl*), cyclooxygenase isoenzyme type 2(COX-2), and phospholipase C zeta (*PLCz*) using real-time quantitative polymerase chain reaction (RT-qPCR). Values carrying different letters (a, b, and c) are significantly different when *p* < 0.05 (n = 8).

**Table 2 life-12-01155-t002:** Evaluation of toxicity by concentration of DMSO in semen cryopreservation.

DMSO (%)	Motility (%)	Progress Motility (%)	VCL (μm/s)	VAP (μm/s)	VSL (μm/s)	Straightness	Linearity (%)	ALH (μm)
0	37.66 ± 5.55 ^a^	7.78 ± 2.76	31.00 ± 1.34 ^a^	18.72 ± 1.05 ^a^	10.06 ± 1.25 ^a^	58.43 ± 4.43 ^a^	38.87 ± 5.15 ^a^	1.15 ± 0.02 ^a^
0.5	38.13 ± 0.32 ^a^	6.91 ± 0.24	23.25 ± 1.76 ^b^	13.66 ± 1.28 ^ab^	4.68 ± 0.89 ^b^	38.72 ± 0.67 ^b^	19.70 ± 3.40 ^b^	0.87 ± 0.08 ^b^
1.0	37.11 ± 0.91^a^	8.26 ± 2.26	21.94 ± 2.78 ^b^	10.04 ± 1.48^b^	4.13 ± 1.37 ^b^	40.11 ± 2.19 ^b^	15.85 ± 2.95 ^b^	0.83 ± 0.75 ^b^
1.5	33.96 ± 1.96 ^ab^	6.06 ± 1.24	25.91 ± 3.44 ^b^	14.09 ± 3.03 ^ab^	5.51 ± 1.82 ^b^	45.96 ± 5.45 ^b^	20.42 ± 5.45 ^b^	0.93 ± 0.78 ^b^
2.0	27.24 ± 1.71 ^b^	3.86 ± 0.48	20.83 ± 2.15 ^b^	11.81 ± 1.18 ^b^	5.55 ± 1.10 ^b^	47.42 ± 11.48 ^b^	26.23 ± 6.95 ^a^	0.88 ± 0.43 ^b^

Curvilinear velocity (VCL), average path velocity (VAP), straight-line velocity (VSL), amplitude of lateral head displacement (ALH). Values with different lowercase superscripts letters (a and b) in the same column differ significantly (*p* < 0.05, n = 3).

**Table 3 life-12-01155-t003:** Assessment of kinematic parameters of quercetin (QRN) for semen cryopreservation.

Groups	Motility (%)	Progress Motility (%)	VCL (µm/s)	VAP (µm/s)	VSL (µm/s)	Straightness (%)	Linearity (%)	ALH (µm)
Control	29.13 ± 0.92 ^b^	14.60 ± 2.73	74.29 ± 5.04 ^a^	37.36 ± 2.90 ^a^	15.76 ± 2.15	41.11 ± 2.37	21.26 ± 1.67 ^a^	2.15 ± 0.10 ^a^
25 µM QRN	30.89 ± 1.15 ^ab^	10.81 ± 1.35	45.73 ± 13.00 ^b^	21.85 ± 2.91 ^b^	8.18 ± 1.60	36.25 ± 4.22	17.24 ± 3.52 ^a^	1.44 ± 0.13 ^b^
50 µM QRN	33.73 ± 0.85 ^a^	16.16 ± 3.25	64.74 ± 10.36 ^a^	30.17 ± 6.64 ^a^	13.40 ± 4.73	44.19 ± 3.56	17.97 ± 2.78 ^a^	1.82 ± 0.21 ^a^
100 µM QRN	28.57 ± 1.02 ^b^	10.89 ± 1.90	62.07 ± 11.30 ^a^	27.80 ± 5.63 ^a^	9.63 ± 1.96	36.48 ± 2.11	13.11 ± 1.26 ^b^	1.84 ± 0.28 ^a^

Values carrying different superscripts (a and b) in the same column are significantly different when *p* < 0.05 (n = 8).

**Table 4 life-12-01155-t004:** Effects of supplementation of quercetin on the post-thaw integrity of plasma membrane, mitochondrial activity, and acrosome integrity of boar sperm.

Groups	HOS (%)	Mitochondrial Activity (%)	Acrosome Integrity (%)
Control	43.1 ± 1.5 ^b^	39.1 ± 0.3 ^c^	66.3 ± 0.8 ^b^
25 µM QRN	45.3 ± 0.6 ^b^	41.9 ± 0.3 ^b^	66.7 ± 1.3 ^b^
50 µM QRN	47.5 ± 0.5 ^ab^	43.0 ± 0.3 ^b^	73.6 ± 1.2 ^a^
100 µM QRN	44.1 ± 0.5 ^b^	43.2 ± 0.6 ^ab^	68.3 ± 1.0 ^b^

a–c, values with different superscript lowercase letters in a column vary significantly (*p* < 0.05, n = 8); HOS, hypo-osmotic swelling; QRN, quercetin.

**Table 5 life-12-01155-t005:** Effects of supplementation of quercetin on the mucus penetration ability of post-thaw boar sperm.

Groups	Number of Sperm Penetrating Mucus	
	1 cm	3 cm
Control	55.0 ± 2.2 ^c^	16.2 ± 1.0 ^b^
25 µM QRN	57.4 ± 1.8 ^b^	18.9 ± 1.1 ^b^
50 µM QRN	66.7 ± 1.9 ^a^	21.9 ± 1.1 ^a^
100 µM QRN	54.3 ± 2.2 ^b^	18.7 ± 0.7 ^b^

Values with different superscripts (a–c) are significantly different when *p* < 0.05 (n = 8).

## Data Availability

The data presented in this study are available on a reasonable request from the corresponding author.

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
