# Peer review of "Cryopreservation of Pig Semen Using a Quercetin-Supplemented Freezing Extender"

_life, 2022, doi:10.3390/life12081155_

Round 1
Reviewer 1 Report
The authors have undertaken a study to examine the cryopreservation of pig semen using a quercetin-supplemented freezing extender. There are some parts in the manuscript which needs major revision and some questions to be answered. The biggest defect of this research is the results part. Significance is misreported in many parts. The analysis of the manuscript is completely wrong.
Sperm dilution rate should be mentioned.
Line 102. In this study, the quercetin has been dissolved in DMSO. For eliminating the effect of DMSO, a treatment containing DMSO must be included.
Line 106. What is the reference which you have selected sperm freezing protocol according to?
Line 114. The calibration of Casa software for pig sperm should be added in the form of a table.
Line 131. Add details of mitochondrial activity measurement.
Line 150. Add ROS measurement details.
Line 155. Added details of apoptosis measurement.
Line 205. Add details of RNA extraction.
The result of this research is written weakly. In the second table, HOST, there are some average differences are reported as “significant” and the same value difference among other treatments, is reported “non-significant”. It is completely wrong!
Mitochondrial activity also appears to be incorrectly reported as significant. Please check it.
In Table 3, capital letters are used for numbers with lower averages. While in other tables it is vice versa.
Table 5. The reported significancy among the treatments is also not correct.
Author Response
The authors have undertaken a study to examine the cryopreservation of pig semen using a quercetin-supplemented freezing extender. There are some parts in the manuscript which needs major revision and some questions to be answered. The biggest defect of this research is the results part. Significance is misreported in many parts. The analysis of the manuscript is completely wrong.
R- We acknowledge the efforts, comments, and suggestions of the reviewer and we considered all these comments and suggestions when revising our manuscript.
Sperm dilution rate should be mentioned.
R- We highlighted this in the text. As you already know, after processing the sperms from pooled semen, the sperms were diluted to a final concentration of 100 × 106 cells/ml and were kept in the straws.
Line 102. In this study, the quercetin has been dissolved in DMSO. For eliminating the effect of DMSO, a treatment containing DMSO must be included.
R- We thank the reviewer for this comment. We would like to draw the reviewer’s attention that the aim of the study was to examine the protective effects of quercetin on freezing/thawing cryodamage effects. We used the same dose of DMSO (1% DMSO, which was proved to be safe according to our preliminary confirmation in Table 2) as a basic cryoprotectant in all groups including the control group. So, the variable is the presence or absence of quercetin.
Line 106. What is the reference which you have selected sperm freezing protocol according to?
R- We thank the reviewer for this question. The experiment was conducted with reference to the method in Kim et al. 2020 as shown in the revised text.
Line 114. The calibration of Casa software for pig sperm should be added in the form of a table.
R- We thank the reviewer for this suggestion. We added a detailed method and attach a picture from the software at the time of analysis. (Please check attached image).
Line 131. Add details of mitochondrial activity measurement.
R- We added detailed methods accordingly.
Line 150. Add ROS measurement details.
R- We have added additional information and references to ROS measurement.
Line 155. Added details of apoptosis measurement.
R- A method for measuring apoptosis has been added to the revised manuscript.
Line 205. Add details of RNA extraction.
R- We wrote the method used to extract RNA.
The result of this research is written weakly. In the second table, HOST, there are some average differences are reported as “significant” and the same value difference among other treatments, is reported “non-significant”. It is completely wrong! Mitochondrial activity also appears to be incorrectly reported as significant. Please check it.
R- We thank the reviewer for the insightful comments. Overall, we confirmed the statistical analysis of the results. Previously, the table gave standard deviations rather than standard errors. This part was corrected, and statistical differences were again shown.
In Table 3, capital letters are used for numbers with lower averages. While in other tables it is vice versa.
R- The content has been corrected.
Table 5. The reported significancy among the treatments is also not correct.
R- The content has been corrected.
Reviewer 2 Report
The present paper investigates the benefits of the flavonoid quercetin on the protection of pig sperm against cold-shock induced by cryopreservation. The paper is very well-written and the objectives supported by a thorough experimental design. The reviewer has no objection regarding the scientific value of the study but the following few points should be observed.
-Do the authors know the advantages of using quercetin vs other antioxidants that have been
successfully tested as supplements in pig sperm freezing medium (Trolox, GSH, lysine, trehalose, vitamin E, tocopherol...)?
-Inclusion of in vivo fertility trials in the same paper would have added interest and increased the
validity of results.
-It is not clear how many ejaculates were included in the study. «Twice a time for 4 weeks from 4
Yorkshire boars». Do you mean four doses were collected twice a week by four weeks, totalling 32 individual doses in eight pools?
-Most boar sperm cryopreservation protocols include programmable freezers because of the
extreme sensitivity of boar sperm to inadequate freezing rates. Also, some protocols concentrate
sperm in the freezing extender to higher numbers (1000 x 106 cells/mL) than the authors did. Please add a reference for the cryopreservation protocol selected.
-You must indicate at which temperature was sperm transported to the lab and maintained before
equilibration at 4ºC.
-Lines 107, 108. Not necessary as they are repeated in lines 110, 111.
-When thawed, was the content of the straws further diluted before assessments?
Author Response
The present paper investigates the benefits of the flavonoid quercetin on the protection of pig sperm against cold-shock induced by cryopreservation. The paper is very well-written and the objectives supported by a thorough experimental design. The reviewer has no objection regarding the scientific value of the study but the following few points should be observed.
R- We acknowledge the efforts, comments, and suggestions of the reviewer and we considered all these comments and suggestions when revising our manuscript.
-Do the authors know the advantages of using quercetin vs other antioxidants that have been successfully tested as supplements in pig sperm freezing medium (Trolox, GSH, lysine, trehalose, vitamin E, tocopherol...)?
R- We thank the reviewer for this excellent comment. Yes, in our lab we published sorts of trials using chemical and biological antioxidants and their effects on canine semen cryopreservation. After testing quercetin on canine sperms freezing (Ref # 29 in the revised manuscript), we tried it on porcine in the current study. We show here our previously published trials:
- Bang S, Qamar AY, Tanga BM, Fang X, Seong G, Nabeel AHT, Yu IJ, Saadeldin IM, Cho J. Quercetin improves the apoptotic index and oxidative stress in post-thaw dog sperm. Environ Sci Poll Res 2022, 29:21925-21934.
- Bang S, Qamar AY, Tanga BM, Fang X, Cho J. Resveratrol supplementation protects against cryodamage in dog post-thaw sperm. J Vet Med Sci. 2021 83(6): 973-980.
- Qamar AY, Fang X, Bang S, Shin ST, Cho J. The effect of astaxanthin supplementation on the post-thaw quality of dog semen. Reprod Dom Anim. 2020; 55(9): 1163–1171.
- Qamar AY, Fang X, Bang SG, Kim MJ, Cho JK. Effects of kinetin supplementation on the post-thaw motility, viability, and structural integrity of dog sperm. Cryobiology, 2020 95: 90-96.
- Qamar AY, Fang X, Kim MJ, Cho JK. Improved viability and fertility of frozen-thawed dog sperm using adipose-derived mesenchymal stem cells. Scientific Reports, 2020 10: 7034.
- Qamar AY, Fang X, Kim MJ, Cho JK. Myoinositol supplementation of freezing medium improves the quality-related parameters of dog sperm. Animals, 2019 9(12): 1038-1049.
- Qamar AY, Fang X, Kim MJ, Cho JK. Improved post-thaw quality of canine semen after treatment with exosomes from conditioned medium of adipose-derived mesenchymal stem cells. Animals, 2019 9(11). 865-877.
-Inclusion of in vivo fertility trials in the same paper would have added interest and increased the validity of results.
R- We totally agree with the reviewer, but at the time of the experiments we were not able to support sows for the in vivo trial through our current fund. We are looking for another funding source to support this work and expand our knowledge in this regard.
-It is not clear how many ejaculates were included in the study. «Twice a time for 4 weeks from 4
Yorkshire boars». Do you mean four doses were collected twice a week by four weeks, totalling 32 individual doses in eight pools?
R- We thank the reviewer for this useful clarification. We edited the text accordingly.
-Most boar sperm cryopreservation protocols include programmable freezers because of the extreme sensitivity of boar sperm to inadequate freezing rates. Also, some protocols concentrate sperm in the freezing extender to higher numbers (1000 x 106 cells/mL) than the authors did. Please add a reference for the cryopreservation protocol selected.
R- We thank the reviewer for this question. The experiment was conducted with reference to the method in Kim et al. 2020 as shown in the revised text.
-You must indicate at which temperature was sperm transported to the lab and maintained before equilibration at 4ºC.
R- We wrote about the temperature brought to the manuscript.
-Lines 107, 108. Not necessary as they are repeated in lines 110, 111.
R- The content has been corrected.
-When thawed, was the content of the straws further diluted before assessments?
R-After thawing, the sperm were diluted to a concentration of 10 × 106/ml and used for analysis. So, we mentioned it.
Reviewer 3 Report
Page 2, line 51. The authors should present the recent reports about redox system and the antioxidant capability of seminal plasma and sperm.
Page 3, line 115. Clarify the kind of counting slide (Leja slide, etc)
Page 4, line 150. Add cellular ROS assay kit (DCFDA)
Conclusions section: Page 11, line 378. The authors should not be absolute in the finding for the 50μM QRN, but they could write that: The optimum concentration of QRN for the protective freeze-thaw pig sperm against OS is 50μM QRN, among the other tested concentrations of QRN.
Author Response
R- We acknowledge the efforts, comments, and suggestions of the reviewer and we considered all these comments and suggestions when revising our manuscript.
Page 2, line 51. The authors should present the recent reports about redox system and the antioxidant capability of seminal plasma and sperm.
R- We thank the reviewer for the useful comment. We added references accordingly.
Page 3, line 115. Clarify the kind of counting slide (Leja slide, etc)
R- We used Standard Count 8 Chamber (Leja ®) for analysis.
Page 4, line 150. Add cellular ROS assay kit (DCFDA)
R- The content has been corrected.
Conclusions section: Page 11, line 378. The authors should not be absolute in the finding for the 50μM QRN, but they could write that: The optimum concentration of QRN for the protective freeze-thaw pig sperm against OS is 50μM QRN, among the other tested concentrations of QRN.
R- We thank the reviewer for the insightful comment. We edited the text accordingly.
Round 2
Reviewer 1 Report
The manuscript is well revised and meets the standard of the journal. It is accepted for publication in this journal.